# Tobacco Mosaic Virus Infection of Chrysanthemums in Thailand: Development of Colorimetric Reverse-Transcription Loop-Mediated Isothermal Amplification (RT–LAMP) Technique for Sensitive and Rapid Detection

**DOI:** 10.3390/plants11141788

**Published:** 2022-07-06

**Authors:** Salit Supakitthanakorn, Kanjana Vichittragoontavorn, Anurag Sunpapao, Kaewalin Kunasakdakul, Pilunthana Thapanapongworakul, On-Uma Ruangwong

**Affiliations:** 1Division of Plant Pathology, Department of Entomology and Plant Pathology, Faculty of Agriculture, Chiang Mai University, Chiang Mai 50200, Thailand; salit_su@cmu.ac.th (S.S.); kaewalin.k@cmu.ac.th (K.K.); pilunthana.t@cmu.ac.th (P.T.); 2Plant Protection Center, Royal Project Foundation, Chiang Mai 50200, Thailand; kan_rpf@yahoo.com; 3Agricultural Innovation and Management Division (Pest Management), Faculty of Natural Resources, Prince of Songkla University, Hatyai, Songkhla 90110, Thailand; anurag.su@psu.ac.th; 4Innovative Agriculture Research Center, Faculty of Agriculture, Chiang Mai University, Chiang Mai 50200, Thailand

**Keywords:** ornamental plant, molecular detection, plant virus, *Tobamovirus*

## Abstract

We detected tobacco mosaic virus (TMV), a member of the genus *Tobamovirus* and one of the most significant plant-infecting viruses, for the first time in a chrysanthemum in Thailand using reverse-transcription polymerase chain reaction (RT–PCR). The TMV-infected chrysanthemum leaves exhibited mosaic symptoms. We conducted a sequence analysis of the coat protein (*CP*) gene and found that the TMV detected in the chrysanthemum had 98% identity with other TMV isolates in GenBank. We carried out bioassays and showed that TMV induced mosaic and stunting symptoms in inoculated chrysanthemums. We observed the rigid rod structure of TMV under a transmission electron microscope (TEM). To enhance the speed and sensitivity of detection, we developed a colorimetric RT loop-mediated isothermal amplification (LAMP) technique. We achieved LAMP detection after 30 min incubation in isothermal conditions at 65 °C, and distinguished the positive results according to the color change from pink to yellow. The sensitivity of the LAMP technique was 1000-fold greater than that of RT–PCR, and we found no cross-reactivity with other viruses or viroids. This is the first reported case of a TMV-infected chrysanthemum in Thailand, and our colorimetric RT–LAMP TMV detection method is the first of its kind.

## 1. Introduction

Chrysanthemums (*Chrysanthemum x morifolium*) are one of the most popular ornamental plants worldwide due to their cultural significance and variations in shape and color [1]. Chrysanthemum plants are susceptible to many phytopathogens, including fungi, bacteria, viruses, viroids, nematodes, and phytoplasmas, with viral diseases being a prominent limiting factor in the chrysanthemum industry [2]. The traditional propagation method for chrysanthemums is vegetative propagation, which is susceptible to viral contamination. The use of virally contaminated mother stocks increases the number of infected plantlets [3]. The meristem tissue culture technique is typically used to decontaminate virus-infected chrysanthemums [4].

More than 10 viruses have been reported to infect chrysanthemums, including the cucumber mosaic virus (CMV), chrysanthemum virus B (CVB), impatient necrotic spot virus (INSV), tobacco mosaic virus (TMV), tomato aspermy virus (TAV), tomato spotted wilt virus (TSWV), tomato leaf curl New Dehli virus (ToLCNDV), turnip mosaic virus (TuMV), and zucchini yellow mosaic virus (ZYMV) [2,4,5,6,7,8,9,10].

TMV, a member of the genus *Tobamovirus* of the family *Virgaviridae*, poses a substantial threat to many tropical and subtropical plants [11]. This virus has a wide host range of more than 100 plant species [12]. TMV is transmitted by mechanical inoculation, grafting, and by many species of aphids in a nonpersistent manner [13]. The genome of TMV comprises approximately 6500 bp of positive-sense single-strand RNA (+ssRNA) [14]. Researchers have reported TMV infection in chrysanthemums but have not yet described any symptoms [9,15,16]. The authors of [17] successfully applied TMV as a vector to express green fluorescent protein (GFP) in chrysanthemums.

Traditionally, researchers have used polymerase chain reaction (PCR) techniques to detect plant viruses and reverse-transcription (RT)–PCR for RNA viruses [18]. However, PCR detection requires expensive, specialized instruments and takes several hours to complete. Recently, researchers have employed loop-mediated isothermal amplification (LAMP) to detect and diagnose plant viruses [18,19,20], which requires only inexpensive equipment [19]. To avoid post-amplification contamination, we used colorimetric RT–LAMP. Positive and negative colorimetric detection results are easily distinguishable thanks to a visible color change, which reduces the complexity and duration of the detection procedure [21]. In this study, we aimed to detect and characterize TMV in Thai chrysanthemums for the first time and to develop a colorimetric RT–LAMP technique to improve the speed, simplicity, and sensitivity of detection.

## 2. Materials and Methods

### 2.1. Sample Collection, RNA Extraction, and cDNA Synthesis

We collected chrysanthemum leaves exhibiting virus-like symptoms, such as mosaic, mottling, chlorosis, vein banding, and malformation, from three cultivation areas in Chiang Mai and one in the Chiang Rai Province of Thailand for detection of TMV using RT–PCR. We extracted total RNA from chrysanthemum leaves using TRIzol^®^ Reagent (Invitrogen, Waltham, MA, USA) according to the manufacturer’s instructions and employed it as a template for the synthesis of complementary DNA (cDNA) using ReverTra Ace^TM^ qPCR RT Master Mix with gDNA Remover (Toyobo, Osaka, Japan) according to the manufacturer’s instructions.

### 2.2. RT–PCR Detection

We performed PCR using EconoTaq^®^ PLUS & PLUS GREEN 2x Master Mixes (Lucigen, Middleton, WI, USA) and two primers specific to the TMV coat protein (*CP*) gene (TMV-spec (5′-CGGTCAGTGCCGAACAAGAA-3′) and Tob-Uni1 (5′-ATTTAAGT GGASGGAAAAVCACT-3′)) [22]. We carried out PCR using a GeneMax Tcs PCR thermal cycler (Bioer, Hangzhou, China) as follows: 94 °C for 4 min, 35 cycles at 94 °C for 30 s, 54 °C for 30 s, 72 °C for 1 min, and 72 °C for 10 min. We analyzed the approximately 657 bp amplicons of the CP gene on 1.0% agarose gel stained with RedSafe^TM^ Nucleic Acid Staining Solution (iNiTron, Gyeonggi, Korea) and visualized them under a DUT-48 Blue Light Transilluminator (MIU Instruments, Hangzhou, China). We calculated the percentage of disease incidence (PDI) according to the formula presented in [23].

### 2.3. Sequence and Phylogenetic Analysis

We purified the PCR product using PCR Clean-Up and Gel Extraction Kits (Bio-Helix, New Taipei City, Taiwan) according to the manufacturer’s instructions and analyzed it using fluorescent dye terminator sequencing with ABI Prism^TM^ 3730xl DNA sequencers (Applied Biosystems, Foster City, CA, USA). We aligned and analyzed the obtained sequence using Molecular Evolutionary Genetics Analysis (MEGA) version X [24] and BLAST analysis, respectively. We deposited the sequence into GenBank to obtain an accession number and calculated the percentage identity matrix compared to TMV isolates retrieved from GenBank using Sequence Demarcation Tool (SDT) version 1.2 [25]. We constructed the phylogenetic tree in MEGA version X, using the maximum likelihood method with 1000 replicates of the bootstrapping value.

### 2.4. Bioassay of Chrysanthemums and Indicator Plants

To confirm the pathogenicity of TMV in chrysanthemums, we examined seedlings (1 month old) of chrysanthemum cultivar Huay Leuk-4. We ground the chrysanthemum leaf that tested positive for TMV with 0.1 M phosphate buffer and used it to mechanically inoculate carborundum-dusted chrysanthemum leaves.

To study the host range and symptom induction, we examined 13 indicator plants. We kept the inoculated plants in a greenhouse at 25–28 °C and observed the symptoms daily. We then subjected all inoculated plants to RT–PCR to confirm the presence of TMV.

### 2.5. Transmission Electron Microscopy (TEM)

To observe the TMV particles under TEM, we prepared the naturally infected chrysanthemum leaves by dipping and negative staining according to the method described in [26]. Briefly, we ground the leaves with 0.1 M phosphate buffer and centrifuged them at 12,000 rpm for 10 min. We deposited the supernatant on a formvar-coated copper grid (300 mesh) that had been glow-discharged. We fixed the grid using 1% glutaraldehyde and stained it with 2% uranyl acetate. We observed the virus under a TEM JEM-2200FS (JOEL, Peabody, MA, USA).

### 2.6. LAMP Primer Design

We designed the LAMP primers (forward inner (FIP), backward inner (BIP), forward (F3), backward (B3), loop-forward (LF), and loop-backward (LB)) using the NEB^®^ LAMP Primer Design Tool version 1.1.0 (New England Biolabs, Ipswich, MA, USA) default setting, based on the TMV *CP* gene sequence retrieved from GenBank (accession no. V01408.1) as a template. All LAMP primers were checked by BLAST to confirm the specificity to TMV.

### 2.7. Optimization of RT–LAMP Conditions

We performed the colorimetric RT–LAMP reaction using WarmStart^®^ Colorimetric LAMP 2x Master Mix with UDG (New England Biolabs, Ipswich, MA, USA). The 25 µL LAMP component contained 12.5 µL WarmStart^®^ Colorimetric LAMP 2x Master Mix with UDG, 2.5 µL LAMP Primer Mix (10×), 1.0 µL target DNA, and 9.0 µL nuclease-free water. We initially tested the LAMP reactions for 60 min incubation at 61, 63, 65, and 67 °C in a dry bath incubator; we then applied the temperature that successfully amplified the LAMP products to determine the optimal incubation duration (30, 45, or 60 min). We analyzed the LAMP products using 1.5% agarose gel electrophoresis (RT–LAMP–AGE), with staining achieved using RedSafe^TM^ Nucleic Acid Staining Solution (iNiTron, Gyeonggi, Korea). Regarding the colorimetric analysis of the RT–LAMP reaction, a color change from pink to yellow after incubation indicated a positive result, whereas no change indicated a negative result.

### 2.8. Sensitivity Assay

To evaluate the limit of detection (LOD) of the RT–LAMP reaction, we prepared cDNA using a 10-fold serial dilution method (10^0^–10^−10^) and measured the amount of cDNA using NanoDrop 2000c (Thermo Fisher, Waltham, MA, USA). We performed RT–LAMP according to the optimal conditions.

### 2.9. Specificity Assay

To determine the possible cross-reactivity with other viruses and viroids, we tested positive cDNA of CMV, CVB, TuMV, chrysanthemum chlorotic mottle viroid (CChMVd), chrysanthemum stunt viroid (CSVd), and cDNA from a healthy chrysanthemum. We performed the RT–LAMP reaction according to the optimal conditions.

## 3. Results

### 3.1. RT–PCR Detection

A total of 110 leaf samples from 13 chrysanthemum cultivars were collected (Table 1). RT–PCR detected TMV (percent disease incidence (PDI) = 0.91%) in 1 out of 110 analyzed samples. The infected sample exhibited a mosaic symptom (Figure 1a). We collected this symptomatic sample from the cultivated area in Chiang Mai Province, but we found no such samples in Chiang Rai Province.

### 3.2. Sequence and Phylogenetic Tree Analysis

We analyzed the nucleotide sequence of the TMV *CP* gene and submitted it to GenBank with the accession no. ON075087. After multiple alignment with 13 other TMV sequences from plants in different countries, including Thai peppers (GenBank accession no. AY633749.1) and Egyptian chrysanthemums (GenBank accession no. GU982315.1), the BLAST analysis and SDT-based color-coded percent identity matrix showed that the *CP* gene of the Thai chrysanthemum TMV shared approximately 95–99% identity with the TMV isolates retrieved from the GenBank database (Figure 2).

The phylogenetic tree analysis demonstrated that the Thai chrysanthemum TMV belonged to a TMV clade closely related to the TMV isolate from China (GenBank accession no. AY566702.1), while the Egyptian chrysanthemum TMV belonged to a nearby branch (Figure 3). The phylogenetic analyses and the host from which the viruses were originally isolated showed that the currently available 36 tobamoviruses were divided into at least eight subgroups according to their host-plant families: Solanaceae-, Brassicaceae-, Cactaceae-, Apocynaceae-, Cucurbitaceae-, Malvaceae-, Leguminosae-, and Passifloraceae-infecting subgroups. TMV was clustered into a Solanaceae-infecting subgroup (Figure 3).

### 3.3. Bioassay of Indicator Plants

We mechanically inoculated 14 indicator plants with TMV and observed two symptom types: local lesions and systemic symptoms. Nine indicator plants exhibited local lesions on inoculated leaves: *Capsicum annum* (chlorotic spots and vein clearing); *Chenopodium amaranticolor* and *C. quinoa* (necrotic spots); *Datura stramonium* and *Nicotiana glutinosa* (necrotic spots that developed into necrosis); and *N. tabacum* cv. *Samsun* NN, *Solanum lycopersicum*, and *Vigna unguiculata* (necrotic spots) (Table 2 and Figure 4). We observed systemic infections in seven indicator plants: *Capsicum annum* (mosaic); *Chrysanthemum x morifolium* (mottling); *C. coronarium* (malformation and yellowing); *N. benthamiana*, *N. tabacum* cv. *Xanthi*, and *Petunia x hybrida* (mosaic and malformation); and *Solanum lycopersicum* (mosaic and vein necrosis) (Table 2 and Figure 4).

### 3.4. Trasmission Electron Microscopy (TEM)

We observed rigid rod particles approximately 200–300 nm in length under TEM in a naturally infected chrysanthemum leaf prepared by dipping and negative staining (Figure 1b).

### 3.5. LAMP Primer Design and Optimization

We designed six LAMP primers: FIP, BIP, F3, B3, LF, and LB (Table 3). We prepared the primer stock concentrations as follows: 16 µM of FIP and BIP, 8 µM of F3 and B3, and 4 µM of LF and LB. We detected a ladder-like pattern in the LAMP products after incubation at 61, 63, and 65 °C for 60 min (Figure 5a). However, we did not detect a LAMP product after incubation at 67 °C. Subsequently, we performed incubation for 30, 45, and 60 min at 65 °C (the manufacturer’s recommended temperature) and detected LAMP products for each incubation duration (Figure 5b). A color change from pink to yellow indicated a positive LAMP result, whereas negative results remained pink (Figure 5a,b). The optimal conditions were incubation at 65 °C for 30 min.

### 3.6. Sensitivity Assay

The limit of detection (LOD) for RT–LAMP by observing the appearance of LAMP products was 10^−8^, which was 10^3^-fold greater than that of RT–PCR (Figure 6a). The LOD for RT–PCR by observing the 657 bp amplicons of the TMV CP gene was 10^−5^ (Figure 6b). The colorimetric observations produced similar results to the gel electrophoresis analysis: the color of the RT–LAMP reactions at 10^0^–10^−8^ changed from pink to yellow, whereas the negative sample and diluted cDNA at 10^−9^–10^−10^ remained pink (Figure 6a).

### 3.7. Specificity Assay

Although the specificity of our LAMP assay to TMV should be tested with other tobamovirus species, any tobamovirus except for TMV was available. Therefore, we verified the binding capacity of LAMP primers in silico. BLAST search found our LAMP primers identical to most TMV strains/isolates but homologous to none of the other tobamovirus sequences. Multiple alignments of partial CP genes of Solanaceae-infecting tobamoviruses demonstrate that our LAMP primers are most unlikely to amplify any gene segment of other tobamoviruses (Appendix A).

To determine the possibility of cross-reaction with other viruses and viroids that could infect chrysanthemums, including CMV, CVB, turnip mosaic virus (TuMV), chrysanthemum chlorotic mottle viroid (CChMVd), and chrysanthemum stunt viroid (CSVd), we performed RT–LAMP according to the optimal conditions with other viruses and viroids. We only detected LAMP products in the TMV lane, which was corroborated by the colorimetric results (Figure 6c), showing that our colorimetric RT–LAMP technique is specific to TMV.

### 3.8. Evaluation of RT–LAMP for Detecting TMV

We used 10 newly collected chrysanthemum samples from 4 cultivars including Huay Leuk-4 (4 samples), New Day (2 samples), Candor Pink (2 samples) and Celebrate (1 sample) and 6 TMV-inoculated indicator plants (chrysanthemum, *Cassia siamea*, *Capsicum annum*, *N. tabacum* cv. *Xanthi*, *Petunia x hybrida*, and *Solanum lycopersicum*) to test the efficiency of colorimetric RT–LAMP. All 10 chrysanthemum samples tested negative. The TMV-inoculated indicator plants exhibiting systemic symptoms (chrysanthemum, *C. annum*, *N. tabacum* cv. *Xanthi*, *Petunia x hybrida*, and *Solanum lycopersicum*) tested positive, whereas the inoculated cassia, which exhibited no symptoms, tested negative (Figure 7a). The colorimetric results corroborated those of RT–LAMP–AGE, with a change from pink to yellow indicating positive samples and no change indicating negative samples (Figure 7b). Using RT–PCR, we observed the 657 bp amplicons of the TMV CP gene in the lanes corresponding to those that contained LAMP products in the colorimetric RT–LAMP analysis (Figure 7c).

## 4. Discussion

TMV has a wide host range, including chrysanthemums. However, based on the scarcity of reports published in the last decade, chrysanthemums do not appear to be a common TMV host [15]. Researchers first detected TMV in chrysanthemums in China, but they provided no description of symptoms [9]. We collected chrysanthemum leaves exhibiting mosaic symptoms for TMV detection using RT–PCR and obtained positive results. We also observed the virus in symptomatic leaves under TEM and confirmed the rigid rod structure unique to *Tobamovirus* [27]. Subsequently, we isolated the virus and maintained it in *N. tabacum* cv. *Xanthi*. To confirm virus pathogenicity, we inoculated chrysanthemums with the virus and observed mottling. The inoculation assay involving indicator plants corroborated the results of previous reports and verified the biological characteristics of chrysanthemum TMV [28].

Sequence and phylogenetic tree analysis confirmed that the virus detected in the chrysanthemums shared 99% identity with the TMV isolates retrieved from GenBank. To confirm a species classification, the nucleotide sequence identity results for a sample must be higher than 90%; a lower score indicates a different species of the same genus [29].

The genus *Tobamovirus* was initially divided into five subgroups according to the amino acid composition and primary structure of coat proteins. Tobamoviruses were originally isolated from host plants in the Solanaceae, Brassicaceae, Cucurbitaceae, Cactaceae and Malvaceae families [30]. Subsequently, the sixth subgroup of the genus *Tobamovirus* was based on tobamoviruses isolated from Passifloraceae after phylogenetic analysis of the 126 kDa and 54 kDa, movement (MP) and CP amino acid sequences, respectively [31]. Presently available the 35 tobamoviruses could be divided further into at least eight subgroups with the addition of Apocynaceae- and Leguminosae-infecting subgroups based on the complete nucleotide sequences and four *Tobamovirus* protein sequences [32].

Researchers have used the RT–LAMP technique to detect many viruses in various plants, such as strawberries and watermelons [33,34]. The RT–LAMP detection of plant viruses requires 30–45 min of incubation, though 15 min is sufficient in some cases [35]. To reduce the duration of the RT–LAMP procedure, the authors of [36] developed a one-step RT–LAMP method for detecting tomato chlorosis virus (ToCV) in plants and whiteflies.

The authors of [37] developed an RT–LAMP technique for TMV detection in tobacco plants after incubation at 65 °C for 60 min and achieved a sensitivity 100 times higher than that of RT–PCR detection. In this study, we developed an RT–LAMP technique that requires 30 min incubation at 65 °C, thus achieving a higher detection efficiency; the sensitivity was also higher and the detection results were easily observable, with a color change from pink to yellow indicating a positive result.

The addition of loop primers to the reaction reduced the DNA amplification time to half or one-third that of the original LAMP method by hybridizing the hairpin loops and extending the region between loops [38]. We used four basic LAMP primers (F3, B3, FIP, and BIP) and two additional loop primers (LF and LB); the previous RT–LAMP technique did not use loop primers [37].

We employed colorimetric RT–LAMP to test newly collected chrysanthemum samples and obtained no positive results, indicating that chrysanthemum TMV infections in Thailand are uncommon. However, to prevent further TMV infections, we should eliminate any TMV host plants to minimize the spread by aphids. TMV constitutes a potential threat to the production of various crops in Thailand, including chrysanthemums, and further research into TMV transmission in the field, TMV-resistant cultivars and TMV-free plantlets, and effective detection methods is necessary to improve the management of this virus.

## 5. Conclusions

Having identified the first reported TMV infection of a chrysanthemum in Thailand, we isolated and characterized the virus according to its biological and molecular properties. We developed a colorimetric RT–LAMP technique with an enhanced detection efficiency that required 30 min incubation at an isothermal temperature of 65 °C. We detected ladder-like patterns of LAMP products that corresponded to the colorimetric results. We easily determined the colorimetric results by observing the color change from pink to yellow, which indicated positive detection. The LOD of colorimetric RT–LAMP was 10^3^-fold that of RT–PCR. The specificity assay found no cross-reactivity with other viruses or viroids. To evaluate the performance of the colorimetric RT–LAMP technique, we tested newly collected chrysanthemum samples and TMV-inoculated indicator plants, which demonstrated effective TMV detection. The colorimetric RT–LAMP method for TMV developed in this study is the first of its kind and can be used for routine detection of TMV due to its speed, accuracy, sensitivity, and specificity.

## Figures and Tables

**Figure 1 plants-11-01788-f001:**
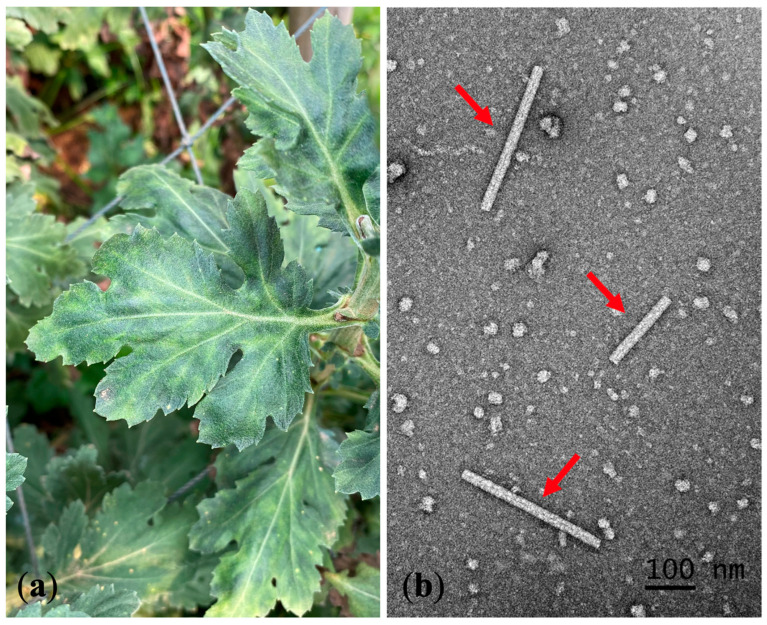
Chrysanthemum leaves exhibiting TMV symptoms and transmission electron micrograph: (**a**) mosaic symptom (dark green area) on naturally infected chrysanthemum leaves; (**b**) transmission electron micrograph of TMV rigid rod structure (red arrows) from a symptomatic chrysanthemum leaf.

**Figure 2 plants-11-01788-f002:**
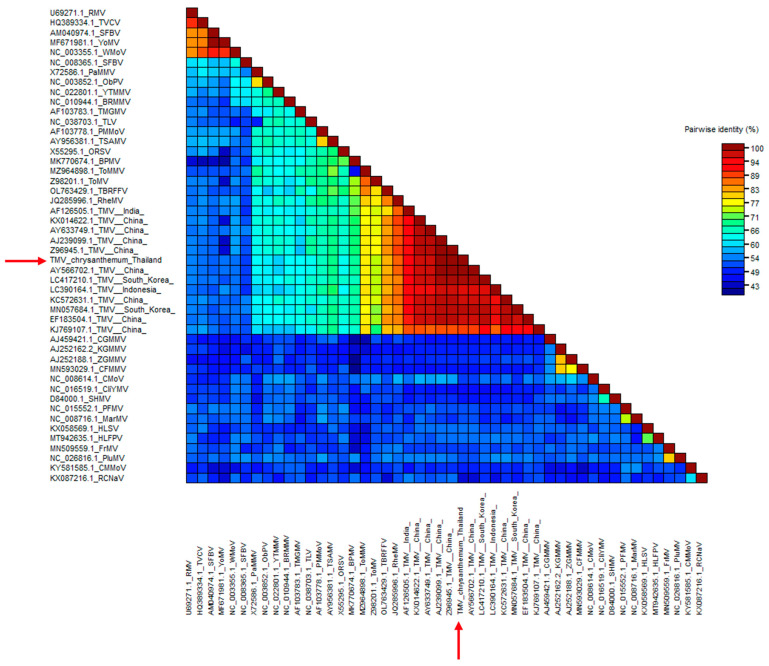
SDT color-coded pairwise identity matrix and phylogenetic tree analysis of TMV CP gene generated from 48 viral sequences comprising 13 TMV sequences and 35 tobamovirus sequences. Color-coded pairwise identity matrix of the Thai chrysanthemum TMV isolate (red arrows). We constructed the pairwise identity matrix using SDT program version 1.2. The cell color indicates the percentage identity score between the two sequences (listed on left and bottom axes) according to the key on the right.

**Figure 3 plants-11-01788-f003:**
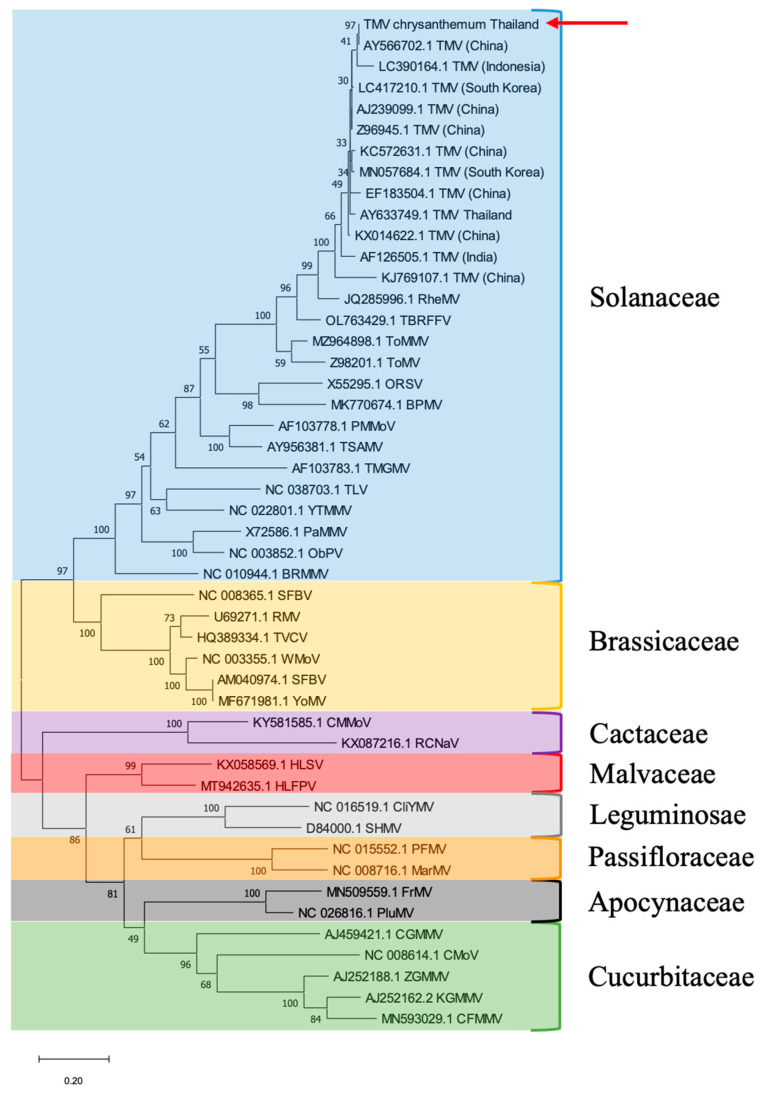
Phylogenetic tree analysis of TMV CP gene generated from 48 viral sequences comprising 13 TMV sequences and 35 tobamoviruses sequences. Thai chrysanthemum TMV is indicated by red arrow. Numbers at internal nodes indicate bootstrap percentages based on 1000 replicates. The scale bar indicates 0.2 substitutions per site. We compared the CP gene sequence of the Thai chrysanthemum TMV with those of other TMV isolates and members of the genus Tobamovirus.

**Figure 4 plants-11-01788-f004:**
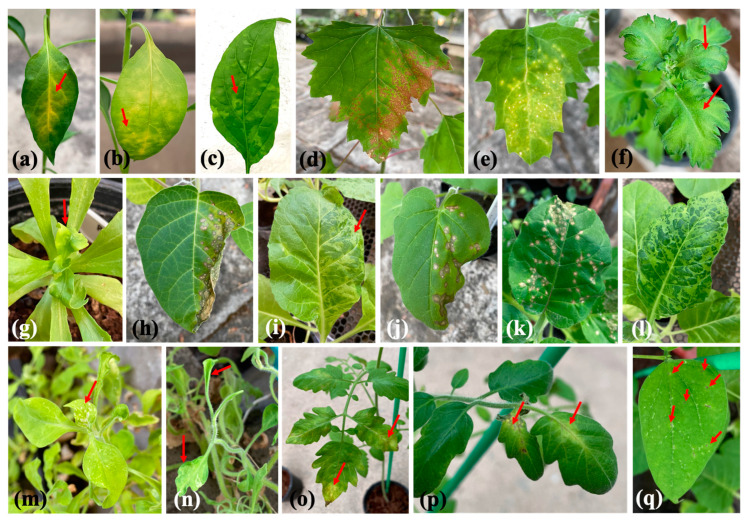
Symptoms exhibited by chrysanthemums and indicator plants (red arrows) mechanically inoculated with TMV: (**a**–**c**) *Capsicum annum*, (**d**) *Chenopodium amaranticolor*, (**e**) *C. quinoa*, (**f**) *Chrysanthemum x morifolium*, (**g**) *Chrysanthemum coronarium*, (**h**) *Datura strmonium*, (**i**) *Nicotiana benthamiana*, (**j**) *N. glutinosa*, (**k**) *N. tabacum* cv. *Samsun* NN, (**l**) *N. tabacum* cv. *Xanthi*, (**m**,**n**) *Petunia* x *hybrida*, (**o**,**p**) *Solanum lycopersicum*, (**q**) *Vigna unguiculata*.

**Figure 5 plants-11-01788-f005:**
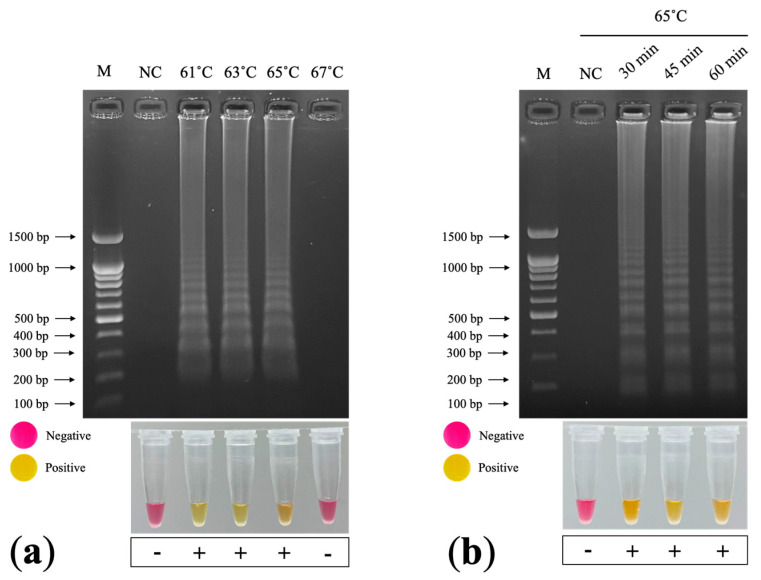
Optimization of colorimetric RT–LAMP TMV detection: (**a**) optimization of temperature 61, 63, 65, and 67 °C for 60 min. (**b**) optimization of incubation time 30, 45, and 60 min at 65 °C. M: 100 bp + 1.5 kb DNA ladder (SibEnzyme, Novosibirsk, Russia); NC: negative control (nuclease-free water).

**Figure 6 plants-11-01788-f006:**
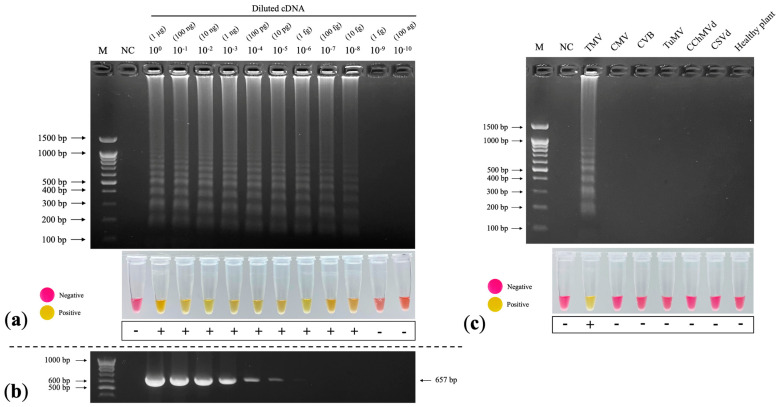
Sensitivity and specificity assays of RT–LAMP TMV detection. (**a**) Limit of detection (LOD) of RT–LAMP–AGE (top) and colorimetric RT–LAMP (bottom) at 10^−^^8^ diluted cDNA. (**b**) LOD of RT–PCR at 10^−^^5^ diluted cDNA. (**c**) Specificity test of colorimetric RT–LAMP for TMV detection, showing a ladder-like pattern of LAMP products in only the TMV lane. M: 100 bp + 1.5 kb DNA ladder (SibEnzyme, Novosibirsk, Russia); NC: negative control (nuclease-free water).

**Figure 7 plants-11-01788-f007:**
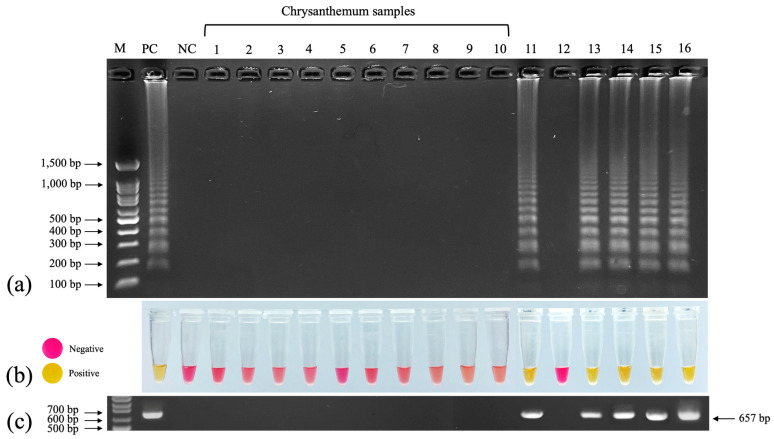
Detection of TMV in plant samples using colorimetric RT–LAMP. Lanes 1–10: chrysanthemum samples from 6 cultivars; lane 11: chrysanthemum inoculated with TMV; lane 12: TMV-inoculated *Cassia siamea*; lane 13: TMV-inoculated *Capsicum annum*; lane 14: TMV-inoculated *Nicotiana tabacum* cv. *Xanthi*; lane 15: TMV-inoculated *Petunia x hybrida*; lane 16: TMV-inoculated *Solanum lycopersicum*. (**a**) RT–LAMP–AGE. (**b**) Colorimetric RT–LAMP. (**c**) RT–PCR with 657 bp amplicons of TMV CP gene. M: 100 bp + 1.5 kb DNA ladder (SibEnzyme, Novosibirsk, Russia); NC: negative control (nuclease-free water).

**Table 1 plants-11-01788-t001:** RT-PCR detection of TMV from 110 samples of 13 chrysanthemum cultivars.

Chrysanthemum Cultivar	No. of Collected and Detected	No. of Positive to TMV	PDI *
Candor Pink	10	0	0
Celebrate	12	0	0
Feeling Green	7	0	0
Huay Leuk-1	10	0	0
Huay Leuk-4	15	1	6.66%
Leopard	9	0	0
New Day	7	0	0
Center	5	0	0
Orange Day	7	0	0
One Way Improve	5	0	0
Vanora	6	0	0
Cornallia	8	0	0
Explorer	9	0	0
**Total**	**110**	**1**	**0.91%**

* PDI: percent of disease incidence.

**Table 2 plants-11-01788-t002:** Symptoms of indicator plants inoculated with TMV.

Indicator Plant	Symptom	
	Inoculated Leaf	Upper Leaf
*Capsicum annum*	CS *, VC	M
*Chenopodium amaranticolor*	NS	-
*Chenopodium quinoa*	NS	-
*Cassia siamea*	-	-
*Chrysanthemum x morifolium*	-	Mo
*Chrysanthemum coronarium*	Y	Ma, Y
*Datura stramonium*	NS, N	-
*Nicotiana benthamiana*	-	M, Ma
*Nicotiana glutinosa*	NS, N	-
*Nicotiana tabacum* cv. *Samsun* NN	NS	-
*Nicotiana tabacum* cv. *Xanthi*	-	M, Ma
*Petunia x hybrida*	-	M, Ma
*Solanum lycopersicum*	NS	M, VN
*Vigna unguiculata*	NS	-

* CS: chlorotic spot, M: mosaic, Ma: malformation, Mo: mottling, N: necrosis, NS: necrotic spot, VC: vein clearing, VN: vein necrosis, Y: yellowing.

**Table 3 plants-11-01788-t003:** LAMP primers used for TMV detection.

Primer	Sequence (5′–3′)	Attachment Position (5′–3′)
F3 *	CCGGAAAAAAGAGTGATGTC	45–64
B3	ACAAGAACACGAACTGAGAT	237–256
FIP	ACTCATTCCTCCAAAATCCTTAACA- AAAGGGAAAAATAGTAGTAGTGATC	F2: 68–92 F1c: 121–145
BIP	AATCGATGATGATTCGGAGGCT- GGAGTAGTGATACTGTAAGACAT	B2: 214–236 B1c: 163–184
LF	TCTTGTTCGGCACTGACC	93–110
LB	GTCGCCGAATCGGATTCGTTTTA	188–210

* F3 = forward primer, B3 = backward primer, FIP = forward inner primer, BIP = backward inner primer, LF = loop-forward primer, LB = loop-backward primer.

## Data Availability

Not applicable.

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
