# Peer review of "Tobacco Mosaic Virus Infection of Chrysanthemums in Thailand: Development of Colorimetric Reverse-Transcription Loop-Mediated Isothermal Amplification (RT–LAMP) Technique for Sensitive and Rapid Detection"

_plants, 2022, doi:10.3390/plants11141788_

Round 1

Reviewer 1 Report

Please find attached my review with comments and suggested amendments.

Overall, I report a lot of minor language errors, so I recommend a revision of English language before publication.

Author Response

Response to reviewer 1

Line 17: we changed “considering” to “considered” as the reviewer’s suggestion.

Line 18: we changed “from” to “in” as the reviewer’s suggestion.

Line 42: we changed “to” to “for” as the reviewer’s suggestion.

Line 47: we changed “tomato yellow leaf curl New Dehli virus (ToYLCNDV)” to “tomato leaf

curl New Dehli virus (ToLCNDV)” as the reviewer’s suggestion.

Line 66: we changed “positive” to “negative” as the reviewer’s suggestion.

Line 73: we changed “detected” to “analyzed” as the reviewer’s suggestion.

Line 74: we added the extended name of PDI whether “percent disease incidence (PDI)” as the reviewer’s suggestion.

Line 84: “this accession number cannot be found in GenBank”

Response: According to the regulation of GenBank, the accession number will be found on the database after publishing the article.

Line 192: “please describe how were these other viruses and viroids chosen? No-target samples for specificity assays have to be genetically related species which infect the same host. please detail how the choice of these species was made”

Response: we added the description whether “with other viruses and viroids that could infect chrysanthemum consisting of CMV, CVB, turnip mosaic virus (TuMV), chrysanthemum chlorotic mottle viroid (CChMVd) and chrysanthemum stunt viroid (CSVd)” in lines 228 – 231 as the reviewer’s suggestion.

Line 280: we deleted the hyphen as the reviewer’s suggestion.

Line 366: we added “time” after the word “first” as the reviewer’s suggestion.

Line 373: we deleted “10 fg/ul which was” in lines 208 and 218 and “(10 pg/ul)” in lines 210 and 219 as the reviewer’s suggestion.

Reviewer 2 Report

This paper is an article titled by “Occurrence of Tobacco Mosaic Virus Infecting Chrysanthemum in Thailand and Development of Colorimetric Reverse Transcription Loop-Mediated Isothermal Amplification (RT-LAMP) for Sensitive and Rapid Detection” by Supakitthanakorn et al. showing that TMV detection level by RT-LAMP was 103 folds higher than that by RT-PCR. Their conclusion seems to be clear. However, the manuscript contains very important problems, flaws and many mistakes and corrections. 1) First of all, they should follow the guideline of this journal. 2) The English language needs vigorous improvements in terms of grammar and colloquial sentences.   3) There is no description on the significant point of their conclusion such as this is the first report in the world, not in their country. 4) The manuscript is biased: only limited number of tobamoviruses, 7 species, were shown in the phylogenetic tree (Figure 2 b). There is no description why these are selected among more than 34 tobamoviruses now (ICTV, Adams et al., 2017).  5) There are no RT-LAMP data of other tobamoviruses using the same primers. 6) One of indicator plants is Nicotiana tabacum cv, Samsun. This must be Samsun NN instead of Samsun because of no systemic symptoms. From these points of view with many other mistakes and corrections, the manuscript is too premature to be submitted to this Journal.

Author Response

Response to reviewer 2

1) First of all, they should follow the guideline of this journal.

Response: we revised and re-checked the manuscript following the guideline of the journal carefully.

2) The English language needs vigorous improvements in terms of grammar and colloquial sentences.

Response: we re-checked and improved the grammar and colloquial sentences in the manuscript as the reviewer’s suggestion.

3) There is no description on the significant point of their conclusion such as this is the first report in the world, not in their country.

Response: This study described the detection of TMV infecting chrysanthemum in Thailand for the first time along with the development of colorimetric RT-LAMP which was highly sensitive to detect TMV. The colorimetric RT-LAMP developed in this study was not the first report in the world but we developed and used the new colorimetric approach with the results which was more sensitive and used less time than the previous one.

4) The manuscript is biased: only limited number of tobamoviruses, 7 species, were shown in the phylogenetic tree (Figure 2 b). There is no description why these are selected among more than 34 tobamoviruses now (ICTV, Adams et al., 2017). 

Response: As the comment of reviewer 3 mentioned that only TMV isolates with one member of tobamoviruses used as the outgroup is sufficient for phylogenetic analysis of only CP gene. We reconstructed the phylogenetic tree as described whether “The nucleotide sequence of CP gene of TMV was analyzed and submitted to GenBank and the obtained accession no. was ON075087. After multiple alignment with other 13 TMV sequences consisting of TMV sequence from pepper of Thailand (GenBank accession no. AY633749.1), TMV sequence from chrysanthemum of Egypt (GenBank accession no. GU982315.1), other TMV sequences from different countries and the sequence of pepper mild mottle virus (PMMoV) (GenBank accession no. AJ242585.1) was used as an outgroup” in lines 84 – 90 along with the new Figure 2.

5) There are no RT-LAMP data of other tobamoviruses using the same primers.

Response: In this study and in our laboratory, TMV is the only Tobamovirus that we have. Therefore, we chose other viruses and viroids consisting of CMV, CVB, TuMV, CChMVd and CSVd which could infect chrysanthemum to test with the RT-LAMP.

6) One of indicator plants is Nicotiana tabacum cv, Samsun. This must be Samsun NN instead of Samsun because of no systemic symptoms.

Response: we added “NN” in lines 140, 169 and Table 1 as the reviewer’s suggestion.

Reviewer 3 Report

Dear Authors,

In your manuscript you identified TMV in chrysanthemum, first time in Thailand in that species. You found one infected case, and in that symptomatic plant you could identify the virus both by TEM and RT-PCR. You also made biotest on different indicators and observed symptoms, specific to TMV. Finally, you made LAMP assay to test TMV presence.

I think that your work is important and would interest readers of MDPI Plants. However, I found that it is not fully and carefully edited and also, you could add some extra information to increase its soundness. The strength of the manuscript is the characterization of the new, chrysanthemum TMV, while I found the LAMP part as its weakness.

Below you will find my comments, suggestions and questions, which, I hope, would help you during the revision of the manuscript, which after a major revision could be in a form what could be accepted in MDPI Plants.

1/ In the 2.1 result section it would be important to detail that 110 samples, from 3 locations were sampled (it is written in the materials and methods, but not detailed). What cultivars were they?  What was the cultivar which was infected? Which samples were used for the LAMP? From the same locations? From the same cultivars? It is also written that chrysanthemum was also included in the biotest. This case should be detail in a different part, as a proof of mechanical transmission of TMV! This is an important message for a vegetatively propagated flower.

2/ First chapter could be the detection of TMV by symptoms, TEM and RT-PCR. Please write more specific chapter titles!

In line 55 is written that TMV was identified infecting chrysanthemum, but it is not mentioned that is there any sequence information about that strain?

As you only had one positive case it wouldn’t be to much work to clone all of the virus. Based on the full genome sequence the phylogenetic analysis would have been more correct. If you keep the only CP analysis, it is not important to cite and include other members of Tobamoviruses for the tree, as it is convincing and not a question that your isolate is a TMV. You can use one Tobamovirus as an outgroup. Based on, it would be interesting to see, why you choose that TMV sequences included to the tree? It would be nice to show some TMV from Thailand or from chrysanthemum is there any available.

3/ in the biotest session I have problems with the numbers. How many indicators were tested? As there was no symptom on Cassia, I won’t list it, only in case if it was RT-PCR positive. But there is no any clue that were theses indicators back tested for the presence of the virus, or not. It would be important to see. As I wrote before, I think the mechanical inoculation on chrysanthemum could be presented separately, or first in this chapter, following with the others.

4/ For the LAMP session, you should first write how the primers were designed, what sequence was used as a reference. I cannot see why a sequences TMV from the Genbank was used for the design and not sequence of your strain?

As you used the colorimetric kit of NEB, you shouldn’t emphasize so many times how the result can be evaluated.

Polymerase for LAMP itself has reverse transcriptase activity, so you don’t have to prepare cDNA for the LAMP test, it works on RNA.

Measuring the concentration of cDNA can be done, but it won’t reflect the concentration of the cDNA as RNA still remains in the mixture. You can simply use dilutions and compared the differences between RT-PCR and LAMP and state how many times is LAMP more sensitive.

It is not written which part of the viral genome was used for LAMP primer design in the results section. I really miss the careful descriptions of the results, please add more information. In the Mat methods it is described that CP is appr 657bp. Was the primers designed to amplify the whole CP? or only part of it?

I don’t see why the specificity of the test was done by using so different viruses?

According to the description in line 204-205 I suspect that Cassia was not successfully infected with the virus.

I would change the panel order on Fig6. First, I would show the LAMP – and as there were faint pink tubes, the panel with the agarose elfo could come, to show that in these suspicious tubes there were not products.

It is possible to carry out LAMP from crude extract of plants. You could easily try this from the indicators, but at least discuss this possibility as something what further simplify the routine TMV test.

In line244 is written that LAMP test for TMV detection was available before. Why did not tried that? What was the main aim to design new primers? And if they were designed why not used the sequence of your strain for that.

Please delete Line258-266, describing the colorimetric LAMP. As it is a commercial kit, you don’t have to explain how to evaluate and how it works.

It is an important part that you collected new samples and tried LAMP on them. However, nothing is mentioned about these new samples. Were they collected from the place where the 1 incidence was found? Form the same cultivar? What was the base for their selection?

As a summary I think that your manuscript could be improved in several ways and I would encourage you to do it so. I would suggest you to ask a native colleague to correct linguistical mistakes.

I do hope that based on my comments you can improve it and its revised form it will be suitable for acceptance in MDPI Plants.

Author Response

Response to reviewer 3

1) In the 2.1 result section it would be important to detail that 110 samples, from 3 locations were sampled (it is written in the materials and methods, but not detailed). What cultivars were they? What was the cultivar which was infected? Which samples were used for the LAMP? From the same locations? From the same cultivars? It is also written that chrysanthemum was also included in the biotest. This case should be detail in a different part, as a proof of mechanical transmission of TMV! This is an important message for a vegetatively propagated flower.

Response:   - What cultivars were they? : More than 10 cultivars of chrysanthemum were collected. They contained both experimentally developed cultivars and commercial cultivars.

                        - What was the cultivar which was infected? :  TMV was detected from chrysanthemum cultivar Huay Leuk-4. This cultivar is a commercial cultivar in Thailand but is not well known in outside the country.

                        - Which samples were used for the LAMP? From the same locations? From the same cultivars? : The sample that was used for LAMP developing is TMV-infected sample of Huay Leuk-4 cultivar. The samples that were used for LAMP evaluating were from Huay Leuk-4 cultivar (4 samples), New Day (2 samples), Candor Pink (2 samples) and Celebrate (1 sample). Also, the previous TMV-infected sample was used as a control. All 10 samples were collected from the same location of the previous TMV-infected sample in Chiang Mai Province.

                        - It is also written that chrysanthemum was also included in the biotest. This case should be detail in a different part, as a proof of mechanical transmission of TMV! : We separated the paragraph of TMV inoculation in chrysanthemum from other indicator plants and Figure 3 was edited by moving chrysanthemum picture as Figure 3A followed by others. The revised section is in lines 131 – 133.

2) First chapter could be the detection of TMV by symptoms, TEM and RT-PCR. Please write more specific chapter titles!

Response: we would like to remain the chapter titles as they were because when we designed the research plan, 1) the samples were surveyed and collected (we did not restrict to detect only TMV at the first time) followed by 2) all samples were subjected to RT-PCR detection and 3) TMV-infected sample was subjected to TEM to confirm and check the presence of TMV. Moreover, reviewers 1 and 3 did not mention about this issue. Therefore, we would like to keep it as the same.

3) In line 55 is written that TMV was identified infecting chrysanthemum, but it is not mentioned that is there any sequence information about that strain?

Response: TMV was previously detected from chrysanthemum in China [ref. 9] and Egypt [ref. 15]. Only TMV infecting chrysanthemum in Egypt had the sequence information. To compare the genetic relationship, we added the sequence of CP gene of TMV from chrysanthemum in Egypt and reconstructed the phylogenetic tree and the results were shown in the section “Sequence and phylogenetic tree analysis” in lines 94 – 102 along with the newly edited Figure 2.

4) As you only had one positive case it wouldn’t be to much work to clone all of the virus. Based on the full genome sequence the phylogenetic analysis would have been more correct. If you keep the only CP analysis, it is not important to cite and include other members of Tobamoviruses for the tree, as it is convincing and not a question that your isolate is a TMV. You can use one Tobamovirus as an outgroup. Based on, it would be interesting to see, why you choose that TMV sequences included to the tree? It would be nice to show some TMV from Thailand or from chrysanthemum is there any available.

Response: we reconstructed the phylogenetic tree as described whether “The nucleotide sequence of CP gene of TMV was analyzed and submitted to GenBank and the obtained accession no. was ON075087. After multiple alignment with other 13 TMV sequences consisting of TMV sequence from pepper of Thailand (GenBank accession no. AY633749.1), TMV sequence from chrysanthemum of Egypt (GenBank accession no. GU982315.1), other TMV sequences from different countries and the sequence of pepper mild mottle virus (PMMoV) (GenBank accession no. AJ242585.1) was used as an outgroup” in lines 84 – 90 along with the newly edited Figure 2.

5)  in the biotest session I have problems with the numbers. How many indicators were tested? As there was no symptom on Cassia, I won’t list it, only in case if it was RT-PCR positive. But there is no any clue that were theses indicators back tested for the presence of the virus, or not. It would be important to see. As I wrote before, I think the mechanical inoculation on chrysanthemum could be presented separately, or first in this chapter, following with the others.

Response:       - How many indicators were tested? : total of 13 indicator plants were tested.

- As there was no symptom on Cassia, I won’t list it, only in case if it was RT-PCR positive : The Cassia that showed no symptom and was negative to both RT-PCR and RT-LAMP detection was not listed as an indicator plant but it was a tested plant.

- But there is no any clue that were theses indicators back tested for the presence of the virus, or not : All tested indicator plants were subjected to RT-PCR to confirm the presence of TMV but the data was not shown. Therefore, we added the sentence “All indicator plants were subjected to RT-PCR for confirmation of the presence of TMV” in lines 369 – 370.

- As I wrote before, I think the mechanical inoculation on chrysanthemum could be presented separately, or first in this chapter, following with the others : We separated the paragraph of TMV inoculation in chrysanthemum from other indicator plants and Figure 3 was edited by moving chrysanthemum picture as Figure 3A followed by others. The revised section is in lines 131 – 133.

6) For the LAMP session, you should first write how the primers were designed, what sequence was used as a reference. I cannot see why a sequences TMV from the Genbank was used for the design and not sequence of your strain?

Response: The information of LAMP primer design and the used TMV sequence were written in the section 4) materials and methods in lines 395 – 399 which was followed to the guideline of the journal.

Not sequence of your strain? : to evaluate the board range for detection of TMV strains, different strain of TMV retrieved from GenBank (GenBank accession no. V01408.1) was used as a template for primer designing and the result showed that LAMP primers designed from different TMV strain could detect TMV chrysanthemum strain in this study.

7) As you used the colorimetric kit of NEB, you shouldn’t emphasize so many times how the result can be evaluated.

Response: we deleted the repetitive sentences about result evaluation in lines 419 and 426 as the reviewer’s suggestion.

8) Polymerase for LAMP itself has reverse transcriptase activity, so you don’t have to prepare cDNA for the LAMP test, it works on RNA.

Response: Thank you for your valuable suggestion. In this study, we synthesized cDNA from extracted RNA because in the comparison of RT-PCR and LAMP, we would like to use the same source of target genomic material.

9) Measuring the concentration of cDNA can be done, but it won’t reflect the concentration of the cDNA as RNA still remains in the mixture. You can simply use dilutions and compared the differences between RT-PCR and LAMP and state how many times is LAMP more sensitive.

Response: we deleted the concentration of cDNA in lines 208, 210, 218, 219 and 436 as the reviewer’s suggestion.

10) It is not written which part of the viral genome was used for LAMP primer design in the results section. I really miss the careful descriptions of the results, please add more information. In the Mat methods it is described that CP is appr 657bp. Was the primers designed to amplify the whole CP? or only part of it?

Response: The LAMP primers were designed based on using whole CP gene sequence of TMV (GenBank accession no. V01408.1), however, when adding the sequence into the program, the program generated many sets of primer automatically and each set of primers were specific to some parts of the sequence not the whole sequence.

11) I don’t see why the specificity of the test was done by using so different viruses?

Response: As the comment of reviewer 1 which asked that “please describe how were these other viruses and viroids chosen? No-target samples for specificity assays have to be genetically related species which infect the same host. please detail how the choice of these species was made” and our response is “other viruses and viroids that could infect chrysanthemum consisting of CMV, CVB, turnip mosaic virus (TuMV), chrysanthemum chlorotic mottle viroid (CChMVd) and chrysanthemum stunt viroid (CSVd)”. As of the comment of reviewer 2 which we responded that “In this study and in our laboratory, TMV is the only Tobamovirus that we have. Therefore, we chose other viruses and viroids consisting of CMV, CVB, TuMV, CChMVd and CSVd which could infect chrysanthemum to test with the RT-LAMP”. Therefore, we thought that the specificity assay using different viruses and viroids was important to confirm there was no cross reactivity.

12) According to the description in line 204-205 I suspect that Cassia was not successfully infected with the virus.

Response: Yes, we agreed. This was the reason why we chose Cassia as one of examined plant samples to evaluate the performance of LAMP compared to other symptomatic indicator plants.

13) I would change the panel order on Fig6. First, I would show the LAMP – and as there were faint pink tubes, the panel with the agarose gel could come, to show that in these suspicious tubes there were not products.

Response: As shown in Figure 6, we decided to show (a) LAMP-AGE firstly followed by (b) colorimetric LAMP and (c) RT-PCR as of the previous LAMP results.

14) It is possible to carry out LAMP from crude extract of plants. You could easily try this from the indicators, but at least discuss this possibility as something what further simplify the routine TMV test.

Response: Thank you for your valuable suggestion.

15) In line244 is written that LAMP test for TMV detection was available before. Why did not tried that? What was the main aim to design new primers? And if they were designed why not used the sequence of your strain for that.

Response:       1) Why did not tried that? What was the main aim to design new primers?

Response: the previously reported TMV LAMP used only four basic LAMP primers without loop primers. The loop primers can accelerate the LAMP detection efficiency and it is difficult to design only loop primers by using the previous LAMP primers sequence. Therefore, we decided to design the new LAMP primer set with loop primers.

2) And if they were designed why not used the sequence of your strain for that.

Response: we selected the TMV sequence in GenBank for designing the new set of LAMP primers not using our sequence because we would like to try the broad range of the primers that were designed from different sequences whether it can detect our TMV or not. The results proved that LAMP primers designed from different TMV sequence can be used to detect TMV in this study effectively.

16) Please delete Line258-266, describing the colorimetric LAMP. As it is a commercial kit, you don’t have to explain how to evaluate and how it works.

Response: we deleted the paragraph describing the colorimetric LAMP evaluation along with the cited references [31-32] as the reviewer’s suggestion.

17) It is an important part that you collected new samples and tried LAMP on them. However, nothing is mentioned about these new samples. Were they collected from the place where the 1 incidence was found? Form the same cultivar? What was the base for their selection?

Response: All 10 new samples were collected from the same cultivation location that we detected the previous TMV and they were the same and also different cultivars. However, the new samples were collected from the different year that we found the TMV. All new samples from the same and different cultivars showed mosaic and mottling symptoms were selected.

Round 2

Reviewer 2 Report

See the uploaded file.

Author Response

Importance!

The manuscript was sent to MDPI English Editing service (ID 45424) along with the additional Layout Editing. The English grammar was edited throughout the manuscript. Moreover, there are few important points which were mainly changed.

  • The title was changed according to the MDPI English Editing.
  • The section of materials and methods was moved prior to the section of result along with the order of the references was edited.
  • The SDT analysis and phylogenetic tree were newly constructed.
  • There 3 new references (30-32) which were added according to the information in discussion section.

*The changed and added words and sentences which were not edited by MDPI English Editing service were red color distributed in the manuscript.

1) First of all, they should follow the guideline of this journal.

Reviewer: I do not think that you follow the guide line of this Journal. If you do, the revised manuscript should be quite similar to recent papers in this Journal. Typically, the form of reference is apparent.

Response: the manuscript was sent to MDPI English Editing service and the optional layout editing was applied. The manuscript was edited as of the guideline. Moreover, we move the section of materials and methods prior the section of results as the suggestion of MDPI English editor.

2) The English language needs vigorous improvements in terms of grammar and colloquial sentences.

Reviewer: They did not realize the corrections. It should be checked by the native expert in this field and the description on the expert needs to add the manuscript.

Response: the manuscript was sent to MDPI English Editing service, the correction of English grammar and other necessary points were edited and improved.

3) There is no description on the significant point of their conclusion such as this is the first report in the world, not in their country.

Reviewer: This conclusion is not so remarkable. More detailed description on the significance of their data. As mentioned below, they have no data on RT-LAMP using other tobamovirus species, it cannot exclude the possibility that their colorimetric RT-LAMP may detect other tobamovirus species which are very close to TMV.

Response: the conclusion was newly revised in the manuscript. As reviewer's mention, when we designed LAMP primers, all primers were checked in BLAST to confirm that the primers were only specific to TMV.

4) The manuscript is biased: only limited number of tobamoviruses, 7 species, were shown in the phylogenetic tree (Figure 2 b). There is no description why these are selected among more than 34 tobamoviruses now (ICTV, Adams et al., 2017).

Reviewer: It seems safe. However it is not clear to use PMMoV as an outgroup. Several more tobamovirus species should be added in the figure, which are very close to TMV in a phylogenetic tree of 34 or more (at present) tobamovirus species. There are neither data of this method on the highly homologous other tobamovirus species nor any variation among 14 TMV isolates in their color and DNA amplification.

Response: The SDT analysis and phylogenetic tree were newly constructed based on the comment of the reviewer using CP gene sequences of other members in the genus Tobamovirus for comparison.

 5) There are no RT-LAMP data of other tobamoviruses using the same primers.

Reviewer: This is wrong from scientific point of view. As a general, the experiments should be done using most critical/important materials regardless of inside/outside the lab after getting those from all over the world. Science is not “easy going”. Otherwise it is of low quality.

Response: As we mentioned above. All primers were checked in BLAST to confirm that the primers were only specific to TMV.

6) One of indicator plants is Nicotiana tabacum cv, Samsun. This must be Samsun NN instead of Samsun because of no systemic symptoms.

Reviewer: They have no data of Samsun NN. The first is to find out the paper of Samsun NN (reference) and then request them for the seeds of both Samsun and Samsun NN. Then inoculation test is needed to do for both. Otherwise there would be a possibility of other named tobacco varieties in the world showing necrotic spots on the inoculated leaves.

Response: we used “Samsun NN” according to the previous comment of the reviewer from the first revision.

Reviewer 3 Report

Many thanks for answering my questions and comments.

The revision was done based on my comments and my questions were answered, I appreciate that. However, the intention of asking about the origin of samples, name of the cultivars was that I missed this information from the manuscript. The authors described me this information, but did not add that information to the manuscript. I would like to suggest to provide this information in the manuscript, in the sample collection section and also in the 1st result section. I would suggest a table format for that. Information about the cultivars would be important to add also when the LAMP test for further samples were described.

Otherwise, I think that the manuscript improved and I believe that after the suggested further revision it will be acceptable in the MDPI Plants.

Author Response

1) The revision was done based on my comments and my questions were answered, I appreciate that. However, the intention of asking about the origin of samples, name of the cultivars was that I missed this information from the manuscript. The authors described me this information, but did not add that information to the manuscript. I would like to suggest to provide this information in the manuscript, in the sample collection section and also in the 1st result section. I would suggest a table format for that.

Response: we added the table (Table 1) mentioned chrysanthemum cultivars used in this study in the manuscript according to the reviewer’s suggestion.
